# Sensor-Fusion-Based Simultaneous Positioning and Vibration Suppression Method for a Three-Degrees-of-Freedom Isolator

**DOI:** 10.3390/mi15030402

**Published:** 2024-03-16

**Authors:** Jing Wang, Lei Wang, Peng Jin, Zhen Zhang, Pengxuan Li, Ritao Xiao

**Affiliations:** School of Instrumentation Science and Engineering, Harbin Institute of Technology, Harbin 150001, China; 16b901030@stu.hit.edu.cn (J.W.); p.jin@hit.edu.cn (P.J.); 17b901033@stu.hit.edu.cn (Z.Z.); pengxuan_li@hit.edu.cn (P.L.); lbx@hit.edu.cn (R.X.)

**Keywords:** vibration suppression, positioning, fusion signal, H∞ controller, 3-DOF isolator

## Abstract

For vibration isolation systems, vibration suppression and platform positioning are both important. Since absolute velocity feedback causes difficulty in achieving positioning while suppressing vibration, an H∞ control strategy based on sensor fusion feedback is proposed in this paper. The signals of inertial and displacement sensors are fused through a pair of complementary filters. Thus, active control based on the fusion signal could concurrently achieve vibration and position control since it is a displacement signal. In addition, the obtained fusion signals have a lower noise level. In this way, simultaneous positioning and vibration suppression can be established using the sensor fusion strategy. On this basis, in order to obtain an optimal H∞ controller, system damping can be maximized by using the performance weight function to attenuate noise; the system bandwidth is determined by the uncertainty weight function, which can avoid the effect of high-frequency modes of the system. The effectiveness of the proposed strategy is verified by comparing it with the conventional absolute velocity feedback strategy on a 3-DOF isolator.

## 1. Introduction

Since the negative stiffness property of an inverted pendulum and the positive stiffness property of a flexible hinge, horizontal isolators based on the inverted pendulum in parallel have a very low system stiffness and excellent low-frequency vibration isolation performance. Therefore, such isolators were often used as pre-isolators in gravitational wave detection to improve the low-frequency isolation performance of the overall isolation system [1,2]. However, vibrations near the natural frequency of the passive system are amplified. In practical applications, inertial sensors are often used to measure absolute velocities to produce skyhook damping, which attenuates vibrations near the natural frequency without affecting the high-frequency attenuation performance [3,4,5].

Currently, many researchers are working on active feedback control methods based on absolute physical quantities in isolation systems [6,7]. With the use of linear quadratic gaussian (LQG) control methods based on velocity feedback, the vibration at the resonance frequency is significantly attenuated for a vibration isolation system [8]. In [9,10], robust Hardy 2/Hardy infinity (H_2_/H∞) control methods were verified to be effective in vibration suppression. An optimal linear quadratic regulator (LQR) controller was designed to reduce tool tip vibrations and improve the machining accuracy of milling robots [11]. An adaptive robust control design was implemented for active suspension systems with uncertainties and hard constraints [12]. An adaptive fuzzy sliding mode control scheme was applied to reduce the seismic response of base-isolated buildings with model uncertainty [13]. In the above studies, the disturbance attenuation capability of the vibration isolation system was improved using an active control method. However, the positioning capability of the system was ignored. For unconstrained systems, drift may occur.

In practical applications, many precision devices require vibration isolation systems with positioning capabilities. For example, in gravitational wave detection, laser interferometers cannot distinguish between arm length variations caused by spatial-temporal distortion of the gravitational wave passing through and those caused by the motion of the test mass. Therefore, vibration isolators suspending a test mass were required to not only attenuate the ground vibration, but also to ensure accurate positioning [14]. In addition, improving the positioning capability of the isolators has two benefits: (1) Reducing the tilt of the vibration isolation platform. Due to the existence of installation errors and manufacturing errors, horizontal vibration isolators based on the inverted pendulum are not completely symmetrical in structure, and there is a coupling between horizontal motion and tilt of the platform [15]. (2) Improved inertial sensor measurement accuracy. The horizontal inertial sensor based on the principle of electromagnetic induction cannot distinguish between horizontal motion and tilt [16,17]. The lower the tilt rate of the platform, the more accurate the horizontal motion measurement result of the inertial sensor. In [18], a sensor fusion strategy was proposed to improve tilt and attenuate vibrations for advanced LIGO. However, the system is costly and complex to design.

In order to simultaneously achieve precision positioning and vibration suppression of vibration isolation systems, hybrid control methods have been investigated [19,20,21,22]. Hybrid adaptive feedforward and feedback controllers have been presented for positioning tracking and vibration suppression [23,24]. In [25], using a double-loop control strategy, an integrated actuator was proven to realize active vibration control during precision positioning. In [26], structural vibrations were suppressed by employing the least mean square (LMS) acceleration feedback algorithm, and parallel control strategies based on proportional–integral (PI) and composite controllers were used to achieve collaborative positioning control. For simultaneous precision positioning and reducing vibrations in flexible spacecraft, a feedback control method integrated with an input shaping technique was considered [27,28,29,30]. All the above control methods can meet the requirements of simultaneously precision positioning and vibration suppression. However, multi-controller strategies or control strategies with input integrators also make these control methods more complex.

For the active control of multi-degrees-of-freedom vibration isolators, the traditional absolute velocity feedback strategy can suppress vibration but result in platform positioning deviations. In order to solve this problem, an H∞ control strategy based on sensor fusion feedback is proposed in this paper. The system decoupling model is first obtained based on the system configuration. Then, in order to obtain feedback signals with better noise performance, a sensor fusion feedback scheme is presented. The resulting fusion signals are displacement signals; therefore, H∞ controllers with integral links can implement positioning and vibration suppression at the same time. In addition, in order to maximize the system damping and reduce the effect of high-frequency modes on the system, a performance weight function is used to limit the system resonance peaks, and an uncertainty weight function is used to determine the system bandwidth. Finally, using a three-degrees-of-freedom (3-DOF) isolator, the effectiveness of the proposed strategy is verified by comparing it with the absolute velocity feedback strategy.

The remaining structures of this paper are organized as follows. Section 2 presents the dynamics model and decoupling control strategy of the 3-DOF isolator. Furthermore, the positioning deviation based on absolute velocity feedback is also displayed in this section. The H∞ control strategy based on sensor fusion is shown, and it is compared with the one based on the absolute velocity feedback in Section 3. The verification and contrast experiments are shown in Section 4. Section 5 provides the conclusions.

## 2. System Model and Positioning Problem Based on the Absolute Velocity Feedback Strategy

### 2.1. Dynamics Model and Active Control Strategy of 3-DOF Isolator

The structural schematic of the 3-DOF isolator is shown in Figure 1. Four inverted pendulums are connected in parallel to support the vibration isolation platform. As shown in Figure 1b, four groups of vibration isolation units are symmetrically distributed on the vibration isolation platform; they are denoted as #1, #2, #3, and #4. Each unit contains a velocity sensor, a displacement sensor, an actuator, and an inverted pendulum. As shown in Figure 1a, the origin of the coordinate system XOY is located at the mass center of the load. The system has three degrees of freedom: horizontal motion along the X and Y axes, noted as longitudinal and sideways motion, respectively; and the third is rotational motion around the Z-axis, which is noted as rotation.

The dynamical equations of the system are
(1)$$Mẍ+\mathrm{Kx}=F_{L}$$
where ***M*** = diag(*m*, *m*, *J*_z_), ***x*** = [*x*, *y*, *γ*]^T^, and ***F***_L_ = [*F*_x_, *F*_y_, *M*_γ_]. *m* and *J*_z_ are the mass and the moment of inertia of the platform, respectively. *x*, *y*, and *γ* are the displacement and angular displacement longitudinally, sideways, and rotationally, respectively. *F*_x_, *F*_y_, and *M*_γ_ are the resultant force and the resultant moment in these three directions, respectively.

The stiffness matrix, ***K***, is
(2)$$K=\left(\begin{array}{ccc} K_{11} & K_{12} & K_{13}\\ K_{21} & K_{22} & K_{23}\\ K_{31} & K_{32} & K_{33} \end{array}\right)$$

Assuming the vibration isolation system is a perfectly symmetrical structure, the stiffness matrix, ***K***, is a diagonal matrix. The elements of the stiffness matrix are
(3)$$\begin{array}{l} K_{11}=k_{x1}+k_{x2}+k_{x3}+k_{x4}\\ K_{22}=k_{y1}+k_{y2}+k_{y3}+k_{y4}\\ K_{33}=r_{k}^{2}(k_{y1}+k_{y2}+k_{y3}+k_{y4})\;+\;r_{k}^{2}(k_{x1}+k_{x2}+k_{x3}+k_{x4}) \end{array}$$
where *k*_x1_, *k*_x2_, *k*_x3_, and *k*_x4_ are the longitudinal stiffness components of the four inverted pendulums. *k*_y1_, *k*_y2_, *k*_y3_, and *k*_y4_ are the sideways stiffness components of the four inverted pendulums. *r*_k_ is the distance from the inverted pendulum to the coordinate axis. The other elements of the stiffness matrix all have zero values. Therefore, there is no coupling between the three degrees of freedom.

According to the configuration of the sensor and actuator in Figure 1b, the relationship between the mass center motion and the sensor measurement is as follows:(4)$$\begin{array}{l} {\left[\begin{array}{cccc} s_{1y} & s_{2x} & s_{3y} & s_{4x} \end{array}\right]}^{T}=T_{s}{\left[\begin{array}{cccc} x & y & \gamma & 0 \end{array}\right]}^{T}\\ {\left[\begin{array}{cccc} v_{1y} & v_{2x} & v_{3y} & v_{4x} \end{array}\right]}^{T}=T_{v}{\left[\begin{array}{cccc} \dot{x} & \dot{y} & \dot{\gamma } & 0 \end{array}\right]}^{T} \end{array}$$

The relationship between the combined force at the center of mass and the actuator force exists as follows:(5)$${\left[\begin{array}{cccc} F_{x} & F_{y} & M_{\gamma } & 0 \end{array}\right]}^{T}=T_{a}{\left[\begin{array}{cccc} f_{1y} & f_{2x} & f_{3y} & f_{4x} \end{array}\right]}^{T}$$
where ***T***_s_ and ***T***_v_ represent the displacement sensor matrix and the velocity sensor matrix, respectively. Their inverses are ***T***_s_^−1^ and ***T***_v_^−1^, respectively. ***T***_a_ represents the actuator matrix; its inverse is denoted as *T*_a_^−1^. The values of ***T***_s_, ***T***_v_, and ***T***_a_ are detailed in Appendix A. *s*_1y_, *s*_2x_, *s*_3y_, and *s*_4x_ are the measurement values of these four displacement sensors, respectively. *v*_1y_, *v*_2x_, *v*_3y_, and *v*_4x_ represent the measurement values of these four velocity sensors, respectively. *f*_1y_, *f*_2x_, *f*_3y_, and *f*_4x_ are the output force of these four actuators, respectively.

Figure 2 shows the decoupling control block diagram of the 3-DOF vibration isolation system, where *C*_x_, *C*_y_, and *C*_γ_ are the longitudinal, the sideways, and the rotational controllers, respectively. Therefore, the controller design for a multi-input and multi-output system is transformed into that for single-input and single-output (SISO) systems.

### 2.2. Positioning Problem Based on Absolute Velocity Feedback Strategy

For the active control of systems, absolute velocity feedback strategies based on inertial sensors are often used to generate skyhook damping. Figure 3 shows the closed-loop control system with absolute velocity feedback, which is affected by external disturbances *w*_1_. *P*_f_ and *G*_geo_ are the transfer functions of the system and inertial sensor, respectively.${\dot{x}}_{p}$ and *x*_p_ represent the platform velocity and the displacement, respectively. *r* is the reference signal. Since *w*_2_ is the introduced disturbance to meet the rank requirement, the weighting factor, *W*_2_, has a very small value of 10^−6^. The weight functions *W*_z3_ and *W*_z4_ are the performance weight function and the uncertainty weight function, respectively. *K* is the H∞ controller. The longitudinal (or sideways) controller with an H∞ norm of 1.0005 is expressed as follows:(6)$$K_{\mathrm{axy}}=\frac{-6766.7(s+572.1)(s+210.4)(s^{2}+4.14s+22.55)}{(s+0.024)(s+5.585\times 10^{4})(s+755.11)(s+62.66)(s+0.40)}$$

Neglecting the high-frequency component that exceeds the bandwidth of the system and the perturbation introduced by the integral term, this controller can be simplified as follows:(7)$$K_{\mathrm{axy}}=\frac{-1.93\times 10^{4}(s^{2}+4.13s+22.55)}{s(s+62.66)(s+040)}$$

The compliance from disturbance input *w*_1_ to platform *x*_p_ is shown in Figure 4. There is a significant amplification of disturbance at low frequencies. There will be a significant drift in the platform when the disturbance, *w*_1_, is a DC signal, as shown in Figure 5. In practical applications, due to the presence of an actuator DC offset, the platform will show significant drift unless its position is limited. Therefore, it is difficult to realize positioning control only using the velocity feedback control strategy. It is necessary to investigate new control strategies to suppress external disturbance and ensure the system positioning capability simultaneously.

## 3. H∞ Control Strategy Based on Sensor Fusion

### 3.1. Active Control System with Sensor Fusion

For a SISO system model, the block diagram of the closed-loop system with sensor fusion feedback is shown in Figure 6. The difference from Figure 3 is that a pair of strict complementary filters consists of a high-pass filter (*H*(s)) and a low-pass filter (*L*(s)) is applied. They are used to fuse the signals from displacement and velocity sensors; the fusion signal is denoted as *y*_p_. The noise characteristics of *y*_p_ are superior to those of either sensor alone over the entire frequency band [18].

The sum of the high-pass filter and the low-pass filter equals 1 at all frequencies in a complex sense, i.e., the phase is 0 and the amplitude is 1.
(8)$$\begin{array}{l} H(s)=\frac{s^{7}+7\omega_{b}s^{6}+21\omega_{b}^{2}s^{5}+35\omega_{b}^{3}s^{4}}{{(s+\omega_{b})}^{7}}\\ L(s)=\frac{35\omega_{b}^{4}s^{3}+21\omega_{b}^{5}s^{2}+7\omega_{b}^{6}s+\omega_{b}^{7}}{{(s+\omega_{b})}^{7}} \end{array}$$
where ω_b_ is the cross-frequency of the two filters.

In the closed-loop system, *u* represents the control signal. The measurement results of the velocity sensor *G*_geo_ need to be calibrated and integrated into displacement signals; its noise is denoted as *n*_1_. *G*_dis_ represents the transfer function of the displacement sensor with noise *n*_2_. The weight functions *W*_z1_ and *W*_z2_ are the performance weight function and uncertainty weight function, respectively. Neglecting sensor noise, the generalized object of the system has four inputs and three outputs, with input ***w*** = [*w*_1_, *w*_2_, *x*_g_, *u*] and output ***z*** = [*z*_1_, *z*_2_, *y*_p_]. Therefore, the generalized object can be represented as follows:(9)$$\begin{array}{l} G(s)=\left[\begin{array}{cc} G_{11}(s) & G_{12}(s)\\ G_{21}(s) & G_{22}(s) \end{array}\right]\\ =\left[\begin{array}{cccc} W_{z1}P_{f}(1/s) & 0 & 0 & W_{z1}P_{f}(1/s)\\ 0 & 0 & 0 & W_{z2}\\ P_{f}(1/s)(H+L) & W_{2} & Lx_{g} & P_{f}(1/s)(H+L) \end{array}\right] \end{array}$$

Disregarding the effect of ground disturbance *x*_g_ due to the relative displacement sensor, the generalized object can be simplified as follows:(10)$$G(s)=\left[\begin{array}{ccc} W_{z1}P_{f}(1/s) & 0 & W_{z1}P_{f}(1/s)\\ 0 & 0 & W_{z2}\\ P_{f}(1/s) & W_{2} & P_{f}(1/s) \end{array}\right]$$

The following is an example of a longitudinal controller design. An integration term should be included in *W*_z1_. In addition, in order to offset the impact of an integrated term on the stability in middle frequencies, *W*_z1_ should have a constant value above 0.2 rad/s. Therefore, the performance weight function *W*_z1_ is
(11)$$W_{z1}=\frac{\rho (s\;+\;0.2)}{(s\;+\;0.001)}$$
where 0.001 is the perturbation to avoid a pole on the imaginary axis and *ρ* is the parameter to be determined in the H∞ optimization design process.

The weight function *W*_z2_ is a limitation on the system bandwidth for eliminating the effect of high-frequency unmodeled dynamics. The bandwidth of the system is required to be no more than 100 rad/s and the closed-loop performance after the bandwidth is attenuated according to −40 dB/dec. In addition, there is a requirement for the ***G***_12_ rank. Therefore, the final form of *W*_z2_ is
(12)$$W_{z2}=\frac{0.01(s+1)(0.01s+1)}{{(0.005s+1)}^{2}}$$

In order to solve the H∞ problem using the DGKF method, the generalized object, ***G***, must satisfy the following assumptions.

**Assumption** **1.*****D***_11_ *must be* 0.

The direct feedthrough from w to z, ***D***_11_, must be zero. Let *P* = *P*_f_ (1/s), and *P* is proper; thus, the assumption is met.

**Assumption** **2.**(***A***
***B***_1_) *is stabilizable and* (***A***
***C***_1_) *is detectable.*

In a well-defined plant, *P* (stabilizable and detectable), since the transfer function of the disturbance w to *y*_p_ contains plant *P*, there are no uncontrollable poles. The assumption is met.

**Assumption** **3.*****D***_12_ *has full column rank.*

The matrix ***D***_12_ denotes the direct feedthrough matrix of ***G***_12_; the transfer function from the control signal, u, to the controlled output, z, is as follows:(13)$$G_{12}=\left[\begin{array}{c} W_{z1}P_{f}\left(1/s\right)\\ W_{z2} \end{array}\right].$$

When one of the matrices has full column rank, the assumption is met. Since *W*_z2_(∞) is a nonzero scalar, the assumption is met.

**Assumption** **4.*****D***_21_ *has full row rank*.

Matrix ***D***_21_ denotes the direct feedthrough matrix of ***G***_21_. Since *W*_2_ is a nonzero scalar, the assumption is met.

According to the above analysis, the system satisfies all the above assumptions. Therefore, the H∞ controller obtained by the DGKF method can make the closed-loop system stable. The value of the H∞ norm is 0.6912 and the corresponding reduced-order H∞ controller is
(14)$$K_{\mathrm{fxy}}=\frac{-9.623\times 10^{6}(s+0.21)(s+14.28)(s+193.2)(s+208.4)}{\left(s+814.4)(s+116.7)(s+0.0074)(s^{2}+155.11s+1.15\times 10^{4}\right)}$$

Neglecting the high-frequency component that exceeds the bandwidth of the system and the perturbation introduced by the integral term, the following reduced-order controller is obtained:(15)$$K_{\mathrm{fxy}}=\frac{-4.08\times 10^{6}(s+0.21)(s+14.28)}{s(s+155.11s+1.15\times 10^{4})}$$

The system models are the same in longitude and sideways. The sideways H∞ controller has the same form. The rotational controller is detailed in Appendix B.

### 3.2. System Performance Analysis

#### 3.2.1. System Stability Analysis

Figure 7 shows the open-loop Bode diagram of the system in these directions. Only the minimum amplitude margin and phase margin are marked in Figure 7. As a comparison, the passive system and the active system with absolute velocity feedback are shown simultaneously. Compared with the passive system, both feedback methods can improve the phase margin of the system in three directions. In addition, the integration link introduced in the fusion feedback system causes the system to lag by 90° around the cutoff frequency, which makes the stability margin of the velocity feedback system slightly superior to that of the fusion feedback system. 

#### 3.2.2. Disturbance Suppression Analysis

Figure 8 illustrates the compliance of the platform displacement, *x*_p_, relative to the disturbance, *w*_1_. From Figure 8, compared with the passive system, the attenuation rate of the closed-loop system with sensor fusion feedback achieves 48.3 dB at natural frequency (0.73 Hz). The proposed strategy is slightly better than the strategy with absolute velocity feedback in longitude and sideways. There is not much difference between the two strategies in rotation. Below the natural frequency, the proposed strategy is better than the strategy with absolute velocity feedback in three directions.

The actuator output force acts at the same position of the system as the disturbance, *w*_1_; the system response with actuator offset is shown in Figure 9. For both the passive system and the absolute velocity feedback strategy, there is a significant drift in the platform. However, there is no significant position change using the proposed strategy.

#### 3.2.3. System Positioning Analysis

Figure 10 shows the transmissibility and step response from the reference input to the platform displacement longitudinally and sideways. As shown in Figure 10a, the closed-loop transfer function with sensor fusion feedback is a low-pass filter. Within the control bandwidth, the transmissibility amplitude is 1. Therefore, the reference signal can be effectively tracked, as shown in Figure 10b. However, for absolute velocity feedback, positioning ability is poor. In rotation, a similar phenomenon can be seen in Figure 11.

## 4. Experiment Verification

### 4.1. Experimental Setup

To verify the effectiveness of the proposed strategy, an experimental setup was built, as shown in Figure 12. The material of the inverted pendulum is 65 Mn steel. The mass of the vibration isolation platform is 20.0 kg, and its material is 304 steel. This vibration isolation system can be divided into three parts: passive vibration isolation structure consisting of four inverted pendulums and the load. The active control unit consists of sensors and actuators. Their location on the platform is shown in Figure 1b. The passive structure and the active control unit form the inverted pendulum active vibration isolation system. The test unit is used to test the system performance; for this purpose, the test 941Bs and the signal analyzer are both calibrated by a third-party organization. The test 961Bs are also located on the platform in a triangular shape, enabling closed-loop control; their measurements are used to obtain the horizontal and rotational angular velocities of the load. Finally, all the test information is displayed in real time on an upper computer. The equipment parameters are shown in Table 1.

Figure 13 illustrates the schematic structure of the active control experimental platform. Inertial and displacement sensors are used to obtain the velocity of the load and the relative displacement between the load and the ground, respectively. Then, the velocity signals are converted into displacement signals by the inertial sensor calibration module. The ADC acquisition module is used to convert the measurements into digital signals, which describe the load’s motion in the longitudinal, sideways, and rotational directions by the sensor matrix, respectively. In the digital control section, the measurements of the above two sensors are fused by the complementary filters, and the fused signals are used as the input of the controller in the three degrees of freedom. Finally, the controller output signals through the actuator matrix are converted to analog signals by the DAC, and the analog signals are converted by the motor into force acting on the system. Therefore, the sensor fusion feedback control is accomplished.

### 4.2. Vibration Suppression Experimental Results

To verify the effect of the proposed sensor fusion feedback strategy, the experiments were implemented in three directions: longitude, sideways, and rotation. The longitudinal system response using the proposed strategy under the floor random excitation is shown in Figure 14. As a comparison, the passive system and the H∞ control strategy using absolute velocity feedback are shown. From Figure 14a, the damping of the passive system is very low. The system resonance peak at 0.74 Hz is significantly suppressed by using both active control strategies. However, concerning the RMS values of the platform, the proposed strategy is slightly better than that with absolute velocity feedback. Similar conclusions can also be obtained in sideways and rotation from Figure 15 and Figure 16.

Figure 17 shows the transmissibility from the floor to the platform using both control strategies in longitudinal and sideways directions. From Figure 17a, there is not much difference between the two control strategies; the system resonance peaks are attenuated by more than 60 dB in the longitudinal direction. From Figure 17b, using the proposed strategy and the one with the absolute velocity feedback, the system resonance peaks are suppressed by 61.6 dB and 38.3 dB, respectively. Therefore, the proposed strategy is superior.

### 4.3. Positioning Experiment Results

Figure 18 illustrates the relative displacement between the platform and the floor when the reference signal is 0. It can be seen that there is a significant drift in the vibration isolation platform using the absolute velocity feedback strategy. The reference signal is efficiently tracked using the proposed strategy.

To further validate the effective positioning capability of the proposed strategy, Figure 19 shows the positioning and vibration isolation results of the platform by using the fusion feedback strategy. The desired displacement is 0.1 mm in longitude and sideways. The desired rotation angle is 1.0 mrad. The longitude and sideways of the platform achieve 95% of the desired orientation displacement around 22.0 s and 21.2 s, respectively. The 95% desired orientation angle is realized around 22.9 s. Moreover, between 20 s and 80 s, the velocity RMSs are 0.30 μm/s, 0.31 μm/s, and 4.51 μrad/s in longitude, sideways, and rotation directions, respectively. In addition, comparisons of the settling times of the positioning and vibration isolation show that the positioning process takes longer. The experimental results demonstrate the feasibility of the proposed strategy in terms of both the vibration isolation and precision positioning for the vibration isolation system.

Figure 20 illustrates the positioning results using the fusion feedback strategy with 0.01 Hz, 0.1 Hz, and 1 Hz sinusoidal desired trajectory, respectively. The peak-to-peak values of the desired sinusoidal signal are 0.2 mm, 0.2 mm, and 1 mrad in longitude, sideways, and rotation. In longitude, the RMSs of tracking error for 0.01 Hz, 0.1 Hz, and 1 Hz are 1.9 μm, 3.2 μm, and 2.9 μm, respectively. The RMSs of sideways tracking error are 1.7 μm, 2.9 μm, and 3.1 μm, respectively. The RMSs of rotational tracking error are 57.3 μrad, 91.6 μrad, and 89.9 μrad, respectively. The experimental results demonstrate the feasibility of the proposed control strategy in terms of accurate positioning for vibration isolation systems.

## 5. Conclusions

Focusing on the platform positioning deviations caused by the absolute velocity feedback strategy, this paper has proposed a novel H∞ control strategy based on sensor fusion feedback. A pair of complementary filters has been used to fuse the signals from inertial and displacement sensors, which reduces the low-frequency noise injection in the feedback control loop. Enabling the proposed strategy, precision positioning and vibration suppression for a 3-DOF isolator can be achieved simultaneously. Finally, simulation and experimental results have verified the effectiveness of the proposed control strategy. In both longitudinal and sideways directions, the ground vibration can be attenuated about 20 dB above 0.4 Hz, and the positioning accuracy (RMS) reaches 1.77 × 10^−5^ m. Tracking error RMSs are both below 3.5 μm for the desired sinusoidal signal in the frequency range of 0.01 Hz to 1 Hz.

## Figures and Tables

**Figure 1 micromachines-15-00402-f001:**
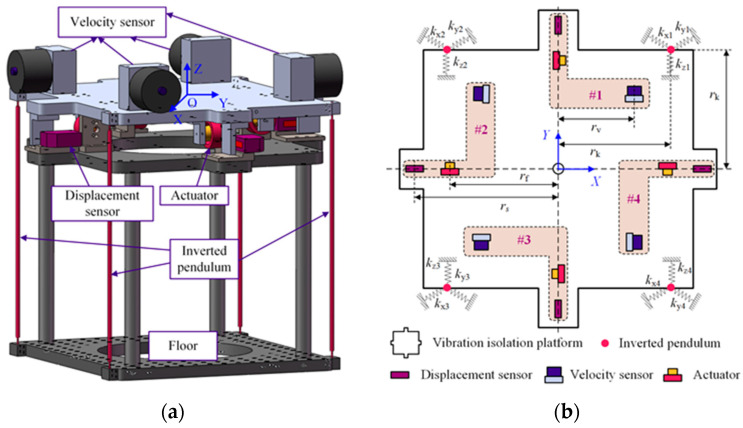
Structure configuration schematic diagram of the 3-DOF isolator. (**a**) CAD model. (**b**) Top view. #1–#4 represent vibration isolation unit numbers.

**Figure 2 micromachines-15-00402-f002:**
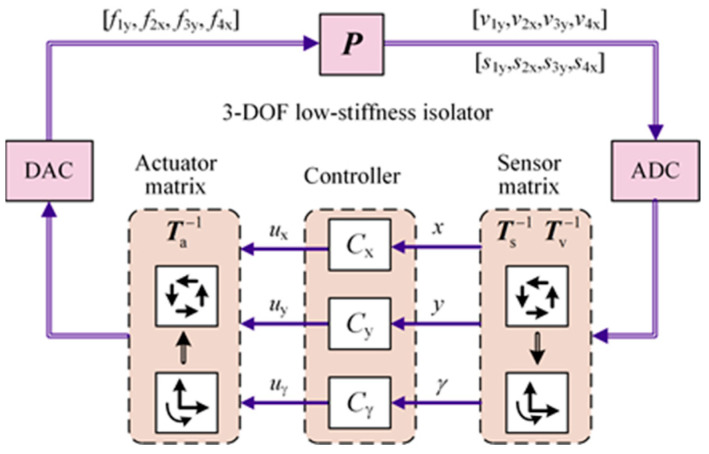
Decoupling control block diagram of the 3-DOF vibration isolation system.

**Figure 3 micromachines-15-00402-f003:**
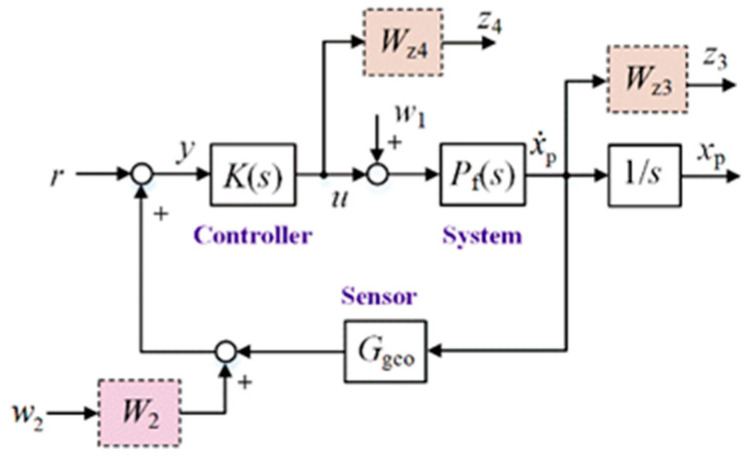
Block diagram of the closed-loop system with absolute velocity feedback.

**Figure 4 micromachines-15-00402-f004:**
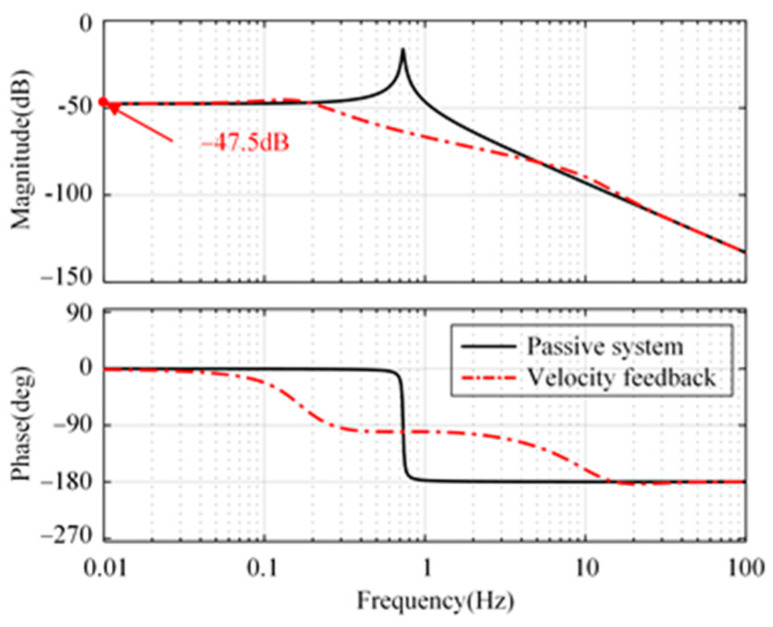
Compliance from disturbance *w*_1_ to platform *x*_p_.

**Figure 5 micromachines-15-00402-f005:**
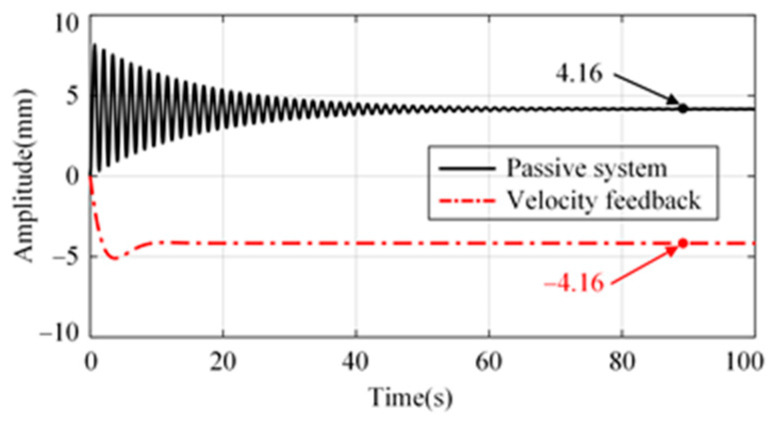
System response when disturbance *w*_1_ is a DC signal.

**Figure 6 micromachines-15-00402-f006:**
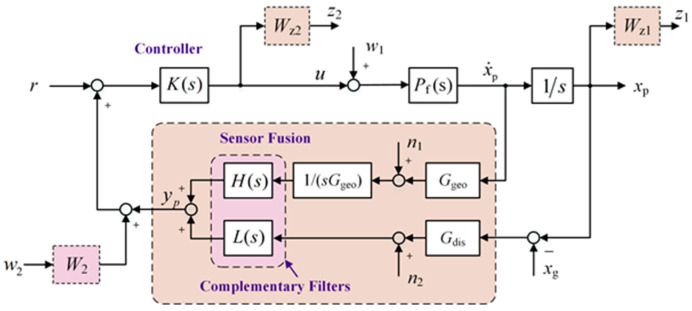
Block diagram of the closed-loop system with sensor fusion feedback.

**Figure 7 micromachines-15-00402-f007:**
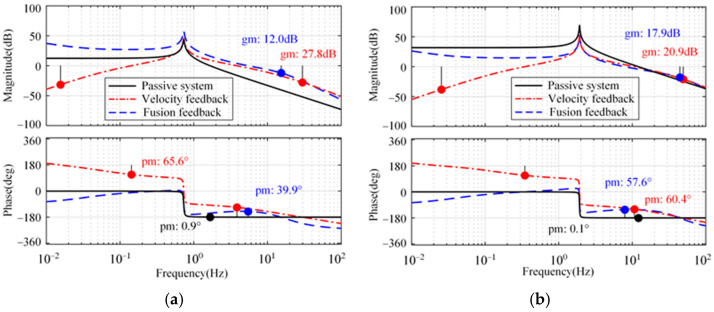
Open-loop Bode diagrams of the system. (**a**) Longitude and sideways. (**b**) Rotation.

**Figure 8 micromachines-15-00402-f008:**
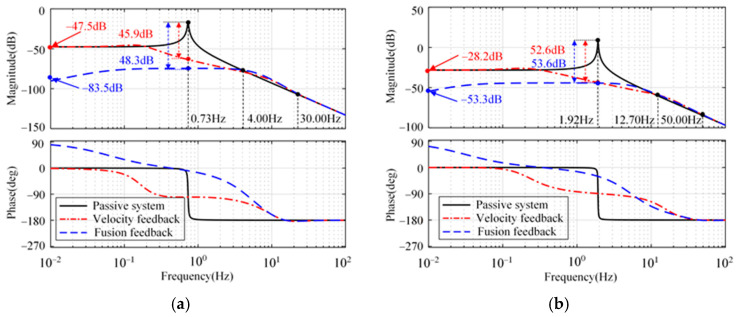
Compliance from the disturbance *w*_1_ to platform displacement *x*_p_. (**a**) Longitude and sideways. (**b**) Rotation.

**Figure 9 micromachines-15-00402-f009:**
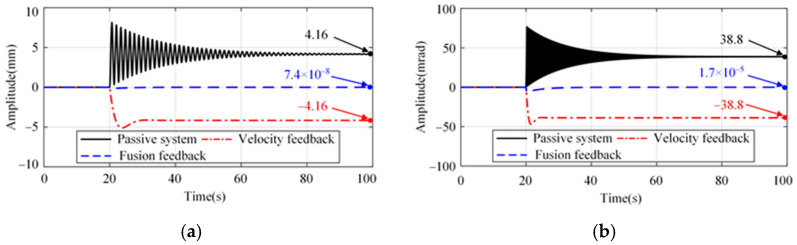
System response with an actuator offset of 1 μN at 20 s. (**a**) Longitude and sideways. (**b**) Rotation.

**Figure 10 micromachines-15-00402-f010:**
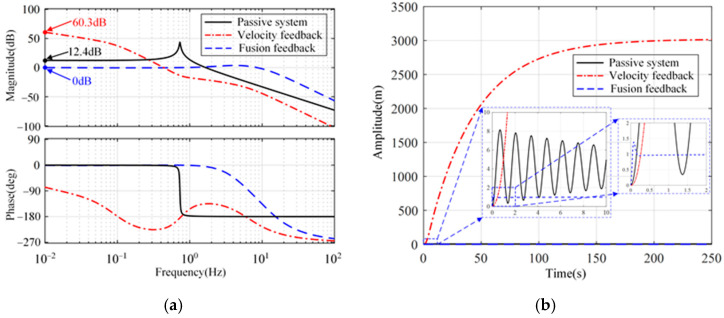
Transmissibility and step response in the longitude and the sideways. (**a**) Transmissibility from the reference input to the platform displacement. (**b**) Step response.

**Figure 11 micromachines-15-00402-f011:**
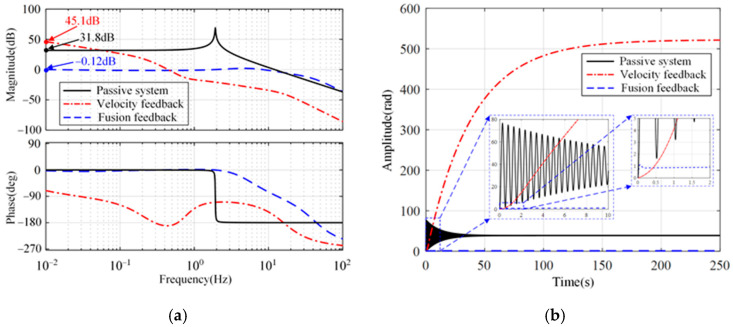
Transmissibility and step response in the rotation. (**a**) Transmissibility from the reference input to the platform displacement. (**b**) Step response.

**Figure 12 micromachines-15-00402-f012:**
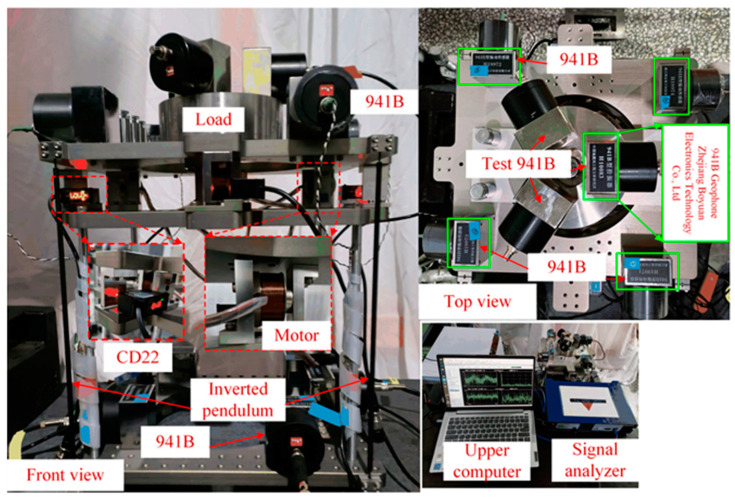
Experimental setup.

**Figure 13 micromachines-15-00402-f013:**
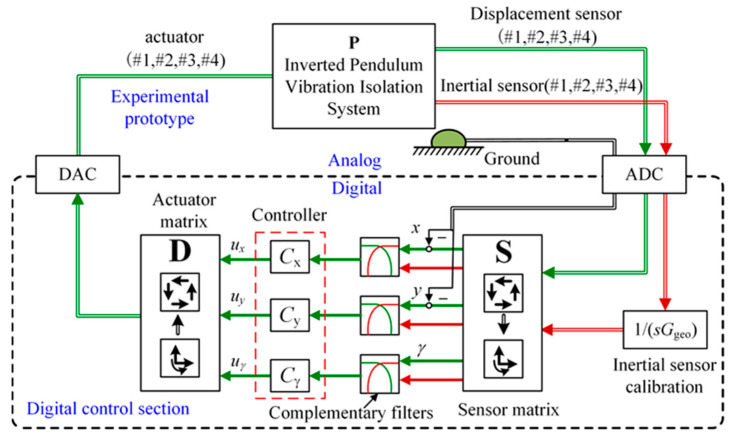
Schematic structure diagram of the active control experimental platform. The red and green lines represent the signals from the inertial and displacement sensors, respectively. The fused signal is the displacement signal, also represented by the green line.

**Figure 14 micromachines-15-00402-f014:**
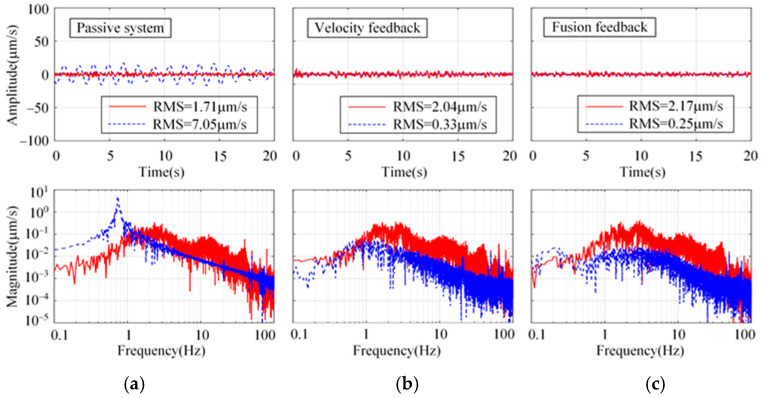
Comparison of the longitudinal system response in time and frequency domains for different control strategies. (**a**) Passive system. (**b**) Absolute velocity feedback. (**c**) Sensor fusion feedback.

**Figure 15 micromachines-15-00402-f015:**
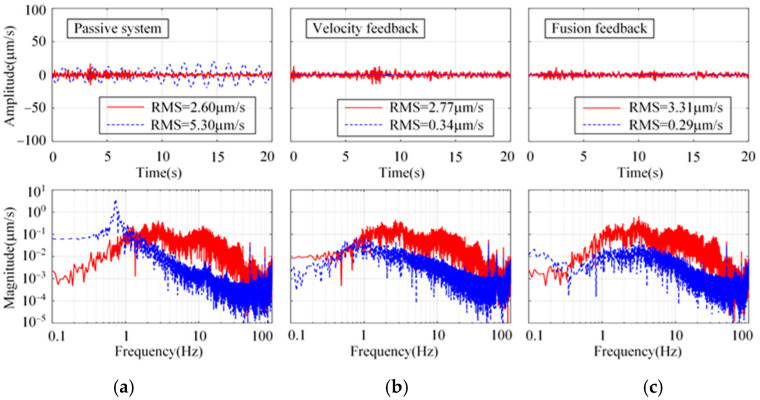
Comparison of the sideways system response in time and frequency domains for different control strategies. (**a**) Passive system. (**b**) Absolute velocity feedback. (**c**) Sensor fusion feedback.

**Figure 16 micromachines-15-00402-f016:**
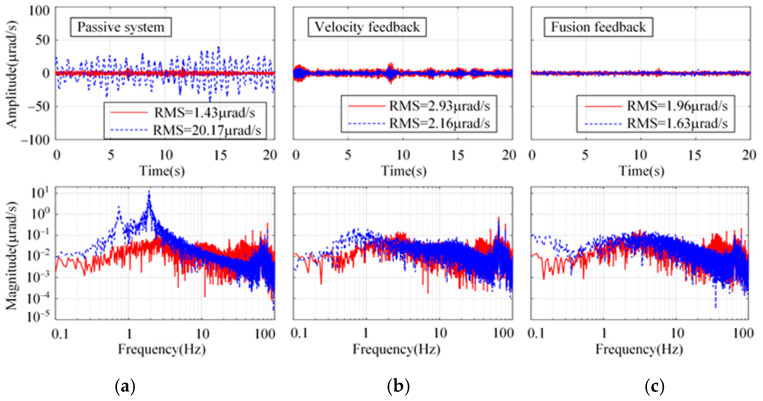
Comparison of rotational system response in time and frequency domains for different control strategies. (**a**) Passive system. (**b**) Absolute velocity feedback. (**c**) Sensor fusion feedback.

**Figure 17 micromachines-15-00402-f017:**
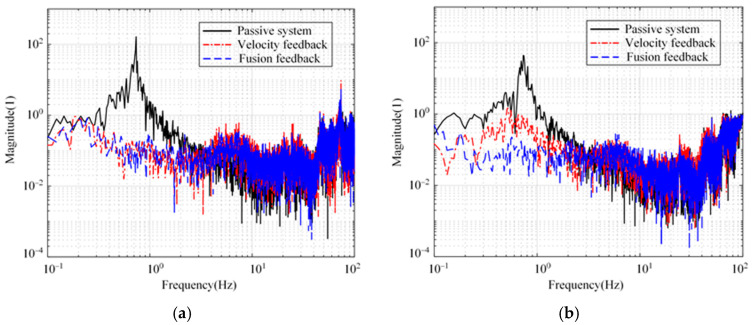
Transmissibility from the floor to the platform for both control strategies. (**a**) Longitude. (**b**) Sideways.

**Figure 18 micromachines-15-00402-f018:**
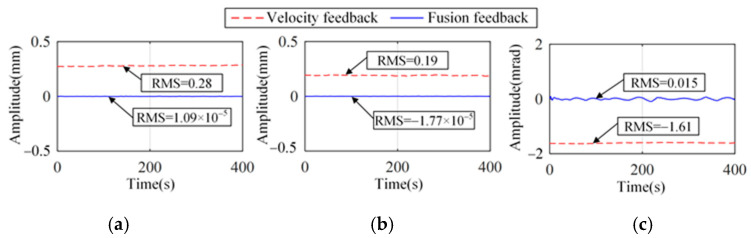
Relative displacement between the platform and the floor for both control strategies. (**a**) Longitude. (**b**) Sideways. (**c**) Rotation.

**Figure 19 micromachines-15-00402-f019:**
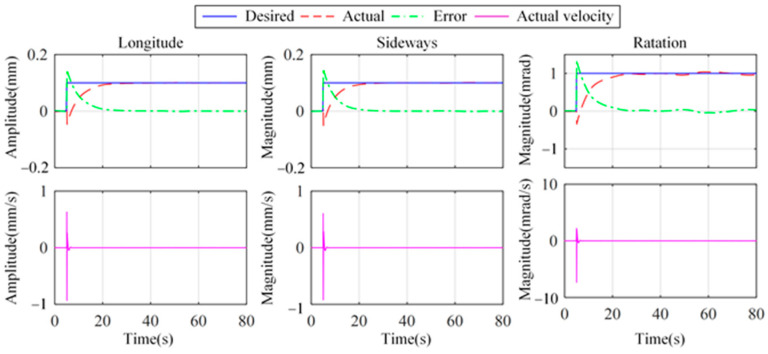
Positioning and vibration isolation results with step desired trajectory at 5 s.

**Figure 20 micromachines-15-00402-f020:**
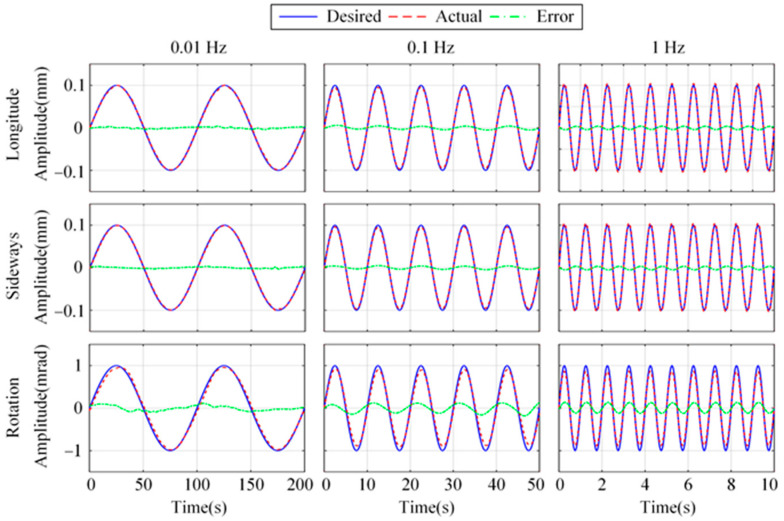
Positioning results with sinusoidal desired trajectory.

**Table 1 micromachines-15-00402-t001:** Equipment parameters.

Equipment	Type	Manufacturer	Parameter	Value
Control board	Self-made PCB	-	-	-
MPU	i.MX RT105	NXP Semiconductors	Main frequency	600 MHz
ADC	AD7606	Analog Devices	Resolution	16 bits
DAC	AD5360	Analog Devices	Resolution	16 bits
Signal analyzer	YSV8016	Beijing Yiyang Strain and Vibration Testing Technology Co., Ltd.	Resolution	24 bits
Displcement sensor	CD22-15	FASTUS	Sensitivity	1 V/mm
Velocity sensor	H941B	Zhejiang Boyuan Electronics Technology Co., Ltd.	Sensitivity	23 m/s^2^
			Natural frequency	1 Hz
Motor	LAC08-004-00A	Beijing Chen Yang Automation Technology	Force constant	1.1 N/m
Motor driver	TA115	Turst Automation, Inc.	Output current	±8A peak
			Bandwidth	5 kHz

## Data Availability

Data are contained within the article.

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
