# Peer review of "Sensor-Fusion-Based Simultaneous Positioning and Vibration Suppression Method for a Three-Degrees-of-Freedom Isolator"

_micromachines, 2024, doi:10.3390/mi15030402_

Round 1
Reviewer 1 Report
Comments and Suggestions for Authors
Manuscript ID: micromachines-2885395
Title: Sensor Fusion Based Simultaneous Positioning and Vibration Suppression Method for a 3-DOF Isolator
This manuscript proposes a new control method called H∞ control to improve the accuracy of platform positioning. This method addresses the limitations of the current absolute velocity feedback strategy by using information from multiple sensors combined through sensor fusion.. The topic is interesting, and the manuscript is well organized. However, the reviewer has some comments below.
1. Abbreviated terms need to be explained. Ex: LQG, H2/ H∞, PI…
2. Check Eq. (1).
3. Enlarge Fig. 12 for visualization.
4. Can the author describe the experiment setup in more detail?
Comments on the Quality of English LanguageMinor editing of English language required
Reviewer 2 Report
Comments and Suggestions for Authors
This paper is devoted to a novel H-infinity control strategy based on sensor fusion feedback.
There is an important remark. As far as I know, DGKF method for synthesizing H-infinity control allows to obtain full-order controller, but not reduced-order controller. The authors of the paper say about the reduced-order controller (see lines 226, 227). This point needs to explain in more details.
Comments on the Quality of English LanguageNo comments.
Reviewer 3 Report
Comments and Suggestions for Authors
- Since the sensor blending method is similar to reference [30], and the H∞ technique also does not appear to have significant differences from the previous method, it is necessary to describe the originality of the paper.
- It would be desirable to express the coordinates of Figure 1 more clearly.
- Provide justification for the H∞ norm of 3.1738.
- Please provide a caption for Figure 6 to include descriptions of n1, n2, and any other omitted elements.
- For Figure 7, it would be beneficial to include an analysis of why velocity feedback is superior to fusion feedback in terms of stability.
- Figure 10(b) step response needs to be more clearly drawn (to show characteristics of fusion feedback).
Comments on the Quality of English Language- Overall, well written, but some unclear expressions need revision.
Round 2
Reviewer 2 Report
Comments and Suggestions for Authors
I recommend this paper for publication.